# Influence of Sputtering Temperature of TiO_2_ Deposited onto Reduced Graphene Oxide Nanosheet as Efficient Photoanodes in Dye-Sensitized Solar Cells

**DOI:** 10.3390/molecules25204852

**Published:** 2020-10-21

**Authors:** Foo Wah Low, Goh Chin Hock, Muhammad Kashif, Nurul Asma Samsudin, Chien Fat Chau, Amaliyah Rohsari Indah Utami, Mohammad Aminul Islam, Cheng Yong Heah, Yun Ming Liew, Chin Wei Lai, Nowshad Amin, Sieh Kiong Tiong

**Affiliations:** 1Department of Electrical & Electronic Engineering, Lee Kong Chian Faculty of Engineering & Science, Universiti Tunku Abdul Rahman, Bandar Sungai Long, Kajang 43000, Selangor, Malaysia; 2Institute of Sustainable Energy, Universiti Tenaga Nasional (The Energy University), Jalan IKRAM-UNITEN, Kajang 43000, Selangor, Malaysia; nurul.asma@uniten.edu.my (N.A.S.); aminbgm@gmail.com (M.A.I.); nowshad@uniten.edu.my (N.A.); siehkiong@uniten.edu.my (S.K.T.); 3School of Electrical & Information Engineering, Tianjin University, 92 Weijin Road, Nankai District, Tianjin 300072, China; kashiff.farooq@gmail.com; 4College of Engineering, Universiti Tenaga Nasional (The Energy University), Jalan IKRAM-UNITEN, Kajang 43000, Selangor, Malaysia; 5Engineering Physics, School of Electrical Engineering, Telkom University, Bandung 40257, Indonesia; amaliyahriu@telkomuniversity.ac.id; 6Department of Electrical Engineering, University of Malaya, Jalan Universiti, Kuala Lumpur 50603, Selangor, Malaysia; 7Faculty of Engineering Technology, Universiti Malaysia Perlis (UniMAP), Sungai Chuchuh, Padang Besar, Kangar 02100, Perlis, Malaysia; cyheah@unimap.edu.my; 8Center of Excellence Geopolymer and Green Technology (CeGeoGTech), School of Materials Engineering, Universiti Malaysia Perlis (UniMAP), Kangar 01000, Perlis, Malaysia; ymliew@unimap.edu.my; 9Nanotechnology and Catalysis Research Centre (NANOCAT), Level 3, Block A, Institute for Advanced Studies (IAS), University of Malaya (UM), Kuala Lumpur 50603, Selangor, Malaysia; cwlai@um.edu.my

**Keywords:** reduced graphene oxide, sputtering temperature, rGO-TiO_2_ nanocomposites, RF sputtering, dye-sensitized solar cells

## Abstract

Renewable solar energy is the key target to reduce fossil fuel consumption, minimize global warming issues, and indirectly minimizes erratic weather patterns. Herein, the authors synthesized an ultrathin reduced graphene oxide (rGO) nanosheet with ~47 nm via an improved Hummer’s method. The TiO_2_ was deposited by RF sputtering onto an rGO nanosheet with a variation of temperature to enhance the photogenerated electron or charge carrier mobility transport for the photoanode component. The morphology, topologies, element composition, crystallinity as well as dye-sensitized solar cells’ (DSSCs) performance were determined accordingly. Based on the results, FTIR spectra revealed presence of Ti-O-C bonds in every rGO-TiO_2_ nanocomposite samples at 800 cm^–1^. Besides, XRD revealed that a broad peak of anatase TiO_2_ was detected at ~25.4° after incorporation with the rGO. Furthermore, it was discovered that sputtering temperature of 120 °C created a desired power conversion energy (PCE) of 7.27% based on the *J-V* plot. Further increase of the sputtering temperature to 160 °C and 200 °C led to excessive TiO_2_ growth on the rGO nanosheet, thus resulting in undesirable charge recombination formed at the photoanode in the DSSC device.

## 1. Introduction

The demand of global energy usage has increased tremendously by 0.9%, equivalent to a 120 million tonnes of oil (Mtoe) in 2019 as compared to 2018 [1]. The consumption of fossil fuel (i.e., oil, coal, gas) is expected to keep rising due to economic growth and increasing population around the world. Further emission of fossil fuels produces carbon monoxide (CO) gas, a driver of the greenhouse effect. This continuous reliance on conventional energy resources will lead to a negative impact on the global warming crisis [2]. These climate change issues will result in erratic patterns like ice melt, sea levels and ocean acidification, plants and animals, and also social effects [3].

In this context, transitioning away from fossil fuels by executing alternative research for renewable energy with low-carbon sources is mandatory. Solar energy is an obvious choice towards a clean energy source, which is free, abundant, and everlasting source that could be provided in a pollution free manner. Nowadays, photovoltaic (PV) technologies have received great attention from researchers due to its ability in generating electricity that is clean, inexpensive, and sustainable, from sunlight [4,5,6]. To date, these technologies are achievable for the optimization of crystalline silicon solar cells at a power conversion energy (PCE) of about 27.6% [7]. Further generation in thin film solar cells involving CIGS, CdTe, and amorphous silicon, could achieve as high as 23.4% in 2019 [7,8,9]. However, these technologies have high cost production and mass scale panels [10].

Emerging PV technology cells of dye-sensitized solar cells (DSSCs), relatively low-cost, and ease for fabrication, have obtained an ideal PCE of 12.6% [11]. Practically, the PCE performance of DSSCs usually depends on the materials used in the photoanode part. Thus, the photoanode is the crucial element, which is applicable for absorbing the incoming light and allowing it to pass into the dyes for photoelectrochemical process [12]. Commonly, titanium dioxide (TiO_2_) is utilized for photoanodes due to its high thin film transparency and good photocatalytic characteristics [13,14]. However, TiO_2_ has some drawbacks such as recombination and the potential of causing undesirable effects for the excited photogenerated electrons in the interfacial transfer and leads to low PCE performance [15].

Recently, a two-dimensional (2D) carbon nanomaterial, graphene, has attracted interest with several outstanding properties that fit the DSSCs and its mechanism features in photoanode [16,17,18]. Furthermore, graphene exhibits efficient charge carrier transport, which will probably facilitate the excited electrons’ flow towards the outer circuit and improve the overall PCE performance of DSSCs [19]. Moreover, graphene has excellent optical transparency properties with good absorption rate that could efficiently allow the illumination light into the dye molecules. However, graphene without functionalized or further incorporation with other metal oxide is insufficient to be applied as a photoanode [20]. Besides, it also suffers from lattice defects and this leads to low PCE for DSSCs [21].

Researchers have attempted to improve the PCE performance of DSSCs by incorporating the TiO_2_ with reduced graphene oxide (rGO) as reported elsewhere [22,23]. Recently, it has been discovered that hydrothermal deposition of rGO could be deposited onto TiO_2_ with various concentrations of GO for photocatalytic degradation of RhB dye [24]. Later, Sayali et al. and their group found that the rGO-TiO_2_ nanocomposite preparation via ultrasound assisted/sonochemical method could obtain good Ti-O-C bonding [25]. Some recent updates about rGO-TiO_2_ formation via different techniques are shown in Table 1. However, these techniques are emphasized on surface deposition/coating and there is a lack of accurate bonding onto the material lattice and inadequate concentration formation by the dopant.

In this paper, the preparation rGO-TiO_2_ nanocomposite as photoanode for DSSCs is reported via an RF sputtering technique approach. Specifically, an optimization of sputtering temperature of the TiO_2_ target and direct penetration of the rGO nanosheet could suppress the recombination while improving the photoinduced charge carrier transport. Furthermore, the sputtering technique is promising to maximize the opportunity to fill the oxygen vacancy to reduce the intrinsic defect of rGO in oxides lattice with TiO_2_. Herein, RF sputtering is a better approach comparable to other physical coating or depositing for exterior dopants. In fact, this technique is associated with a better adhesion and uniform distribution onto rGO nanosheet with efficient atom bombardment. Until now, detailed studies of rGO decorated with TiO_2_ with various sputtering temperatures onto rGO nanosheet for DSSCs performance are still lacking. Yet, the influence of sputtering temperature of TiO_2_ onto rGO nanosheet, reaction mechanism, and their physical/chemical characteristics as photoanode remains unclear. Henceforth, comprehensive work is conducted to optimize the rGO-TiO_2_ nanocomposite as photoanode element for DSSCs and to be tested under 100 W solar illumination power.

## 2. Experimental Details

### 2.1. Materials

Graphite powder (<20 µm; 99.99%); potassium permanganate, KMnO_4_ (≥99.0%); hydrazine solution (35 wt% in H_2_O); fluorine doped tin oxide coated glass slide, FTO coated glass (surface resistivity: ~7 Ω/sq), Di-tetrabutylammonium cis-bis(isothiocyanato)bis(2,2′-bipyridyl-4,4′-dicarboxylato)ruthenium(II) (ruthenium dye), platinum, Pt (≥99.9% trace metals basis), and silver conductive paste were purchased from Sigma Aldrich, Malaysia. Sulfuric acid, H_2_SO_4_ (95–97%); ortho-phosphoric acid, H_3_PO_4_ (85%); hydrogen peroxide, H_2_O_2_ (30%); hydrochloric acid fuming, HCl (37.0%); absolute ethanol, C_2_H_5_OH (≥99.5%), acetonitrile, C_2_H_3_N (41.05 g/mol), and potassium iodide electrolyte, KI (≥99.0%) were purchased from Merck, Malaysia. Titanium target for sputtering (99.99% purity, diameter in 50,800 µm with thickness of 6350 µm) was purchased from ULVAC Inc. The deposition process of TiO_2_ onto rGO nanosheet was conducted using an RF sputtering machine at SIRIM Berhad, Malaysia.

### 2.2. GO and rGO Preparation

Ideal GO and rGO nanosheets were synthesized via improved Hummer’s method and chemical reduction technique as reported in our previous work [31,32,33]. The overall reaction is illustrated in Figure 1a whereas the chemical structure of graphite, GO, and rGO are shown in Figure 1b–d, respectively. Comprehensively, GO was prepared from graphite powder as the precursor material via improved Hummer’s method. A total of 1.5 g of graphite powder was poured into an acid ratio of 9:1 (H_2_SO_4_:H_3_PO_4_) [34]. Next, 9.0 g of oxidizing agent, KMnO_4_, was then slowly poured into the mixture under ice bath condition (<20 °C). The solvent color changed from dark purplish green to dark brown, indicating that the oxidizing process was taking place. After 24 h, the solvent mixture was slowly transferred into ~200 mL ice solution and the overall reaction was conducted under ice bath condition. The oxidization process was terminated by adding 3 mL of H_2_O_2_ dropwise into the mixture and turned the color from dark brown to light brownish, indicating that a high oxidation level of graphite was well formed [34]. The suspension was centrifuged and washed with diluted HCl and DI water until pH7 was achieved. The sol-gel GO byproduct was formed after being dried for 24 h in a dry oven. Furthermore, 1.26 g of fine GO was produced from graphite powder. For rGO synthesis, it was well prepared via a chemically reduction process. Additionally, 300 mg of GO flakes were added into 100 mL distilled water while 100 µL of hydrazine solvent was immediately dropped into the mixture. The overall reaction was heated under oil bath conditions and maintained at ~80 °C [35]. The mixture was centrifuged and the supernatant was decanted away. Lastly, approximately 0.84 g of rGO samples were formed after being dried in a dry oven for 24 h. Henceforth, the yield production of synthesized rGO from GO went up to 67%.

### 2.3. rGO-TiO_2_ Nanocomposite Formation

The rGO-TiO_2_ nanocomposite via sputtering technique was prepared as an efficient photoanode for DSSCs devices as depicted in Figure 2. Firstly, the rGO nanosheet layer was deposited onto FTO glass via an electrodeposition technique as reported in our finding [33]. Size of the entire DSSCs device had been fixed with 2 cm × 2 cm area for both the anode and the cathode. The rGO was deposited on FTO glass for the anode part with an active area of 0.67 cm^2^. From our understanding, the sputtering method is one of the effective routes to produce photoanode to achieve an ideal PCE of DSSC performance [36]. In other words, the sputtering technique has the potential to allow more dopant atom particles to penetrate onto the rGO nanosheet under high acceleration and are well formed within a second [37]. Thus, it would enhance the properties of rGO-TiO_2_ nanocomposites in terms of charge carrier transport rate, resulting in high PCE of DSSC performance. In this typical procedure, several FTO with coated rGO were placed for RF sputtering with different sputtering temperatures of 40, 80, 120, 160, and 200 °C. The utilized titanium dioxide target was placed in a chamber with the optimization of being placed with distance of 10 cm apart [33]. For the uniformity of dopant onto the rGO nanosheet, the sputtering duration of TiO_2_ and input power were maintained at 60 s and 150 W, respectively. The flow rate of Argon, Ar, gas was 15 mL/min, pressure at 266.64 mPa with base pressure of 0.67 mPa. Finally, the rGO-TiO_2_ nanocomposite was successfully formed for the photoanode element.

### 2.4. DSSCs Fabrication 

Theoretically, a working DSSCs device is integrated in a sandwich configuration, which consists of TCO/photoanodes/dye/electrolyte/counter electrode/TCO as shown in Figure 3. Practically, our study aims at modification of rGO photoanodes (conventional in TiO_2_ material), which is the core element for the incoming light absorption ability. The main role of the photoanode is used to allow the excited photo-electron from dye molecules into the conduction band of TiO_2_ under the illumination process. In addition, the incorporation of rGO with TiO_2_ is applied to lift-down the incoming electrons at TiO_2_ since it is a wide band gap metal oxide semiconductor material (3.2 eV). The role of rGO also helps in the internal movement of exciton electrons from the valence band of TiO_2_ into the conduction band of TiO_2_, where it is possesses high carrier mobility with almost zero band gap characteristics [16]. In this way, the rGO-TiO_2_ nanocomposite could efficiently transfer the excited photo-electrons by minimizing the charge recombination.

For details of DSSCs fabrication process, photoanode contained rGO-TiO_2_ was soaked into a solvent containing 0.5 mL N719 dye (0.3 mM) and C_2_H_5_OH for 24 h. The photoanode was rinsed with C_2_H_3_N and post baked for 10 min. Then, the counter electrode was coated with Pt with an active area of 0.67 cm^2^ via spin coating method. Both of the electrodes were sandwiched and 0.5 M KI electrolyte were dropped on the gap. The overall device was sealed by silver paste.

### 2.5. Characterization

The surface morphologies of graphite, GO, and rGO were observed using field emission scanning electron microscopy (FESEM, FEI Quanta 200 FEG) with attachment of energy dispersive X-ray analysis (EDX), 5 kV. For surface morphologies of TiO_2_, it was viewed under scanning electron microscopy TM3030 tabletop microscope at a working distance of approximately 2.0 mm at high vacuum mode with 5.0 kV. Besides, the rGO-TiO_2_ nanocomposite was monitored under HITACHI UHR Cold-Emission FE-SEM SU 8000. The lattice of rGO-TiO_2_ nanocomposite was examined under high-resolution transmission electron microscopy (HRTEM), JEM 2100F with an accelerating voltage of 200 kV. The topologies of the rGO-TiO_2_ nanocomposite were measured using atomic force microscope controller—AFM5000II with 3D rotation. The purity phases of TiO_2_, crystalline of rGO, and rGO-TiO_2_ nanocomposite were determined using X-ray diffraction (XRD), D8 Advance X-ray diffractometer-Bruker AXS, the spectra were measured from 10° to 70° with scanning rate of 0.033 deg/s under CuKα radiation (λ = 1.5418 Å). The structural characterization of rGO, TiO_2_, and rGO-TiO_2_ nanocomposite were recognized by the Raman analysis, Renishaw inVia microscope with applied HeCd laser source with λ = 514.0 nm at room temperature. Furthermore, its functional groups of rGO, TiO_2_, and rGO-TiO_2_ nanocomposite were identified under the Fourier transform infrared (FTIR) spectroscopy, Bruker-IFS 66/S along 500–4000 cm^−1^ wavelength by the KBr pellet method. The *J-V* curves of DSSCs were obtained from Autolab PGSTAT204 with solar irradiation (mercury xenon lamp) under 100 W input power.

## 3. Results and Discussion

### 3.1. Morphology 

The FESEM images of graphite, graphene oxide (GO), and rGO are shown in Figure 4a–c, respectively. There are thick massive graphite flakes with nonuniform graphitic sheets distributed along the sample (Figure 4a). Figure 4b shows thin layers of GO after oxidation and exfoliation. On the other hand, Figure 4d shows the TiO_2_ nanoparticles, which are sputtered on the surface of rGO to form the rGO-TiO_2_ nanocomposite (Figure 4e) with average nanoparticles of ~30 nm (inserted in Figure 4e). Furthermore, the EDX results revealed that the atomic, at.%, content of carbon, C, element in the rGO-TiO_2_ nanocomposite have been recorded at 37.29%, which is almost three times more than the titanium, Ti, element with 12.61%. However, the overall oxygen, O, remained the most contained element due to its contribution from the oxygenated group of graphene and also oxygen from TiO_2_. The broad peak detected at 4.5 eV with high Ti content, is mainly due to the huge amount of Ti material sputtered and mixed along the rGO surface.

### 3.2. HRTEM

Further insight into the detailed microstructure of HRTEM and the typical image under 2 nm magnification determined that the TiO_2_ is homogenously well anchored with rGO and formed rGO-TiO_2_ nanocomposite as shown in Figure 5. Generally, the brighter color (0.336 nm) represented rGO nanomaterial; darker color (0.349 nm) those composed of TiO_2_ nanoparticles, while grey color (0.399 nm) denoted the rGO-TiO_2_ nanocomposite [33]. These phenomena were in agreement as Ti-O-C bonding, which was present and proven in Figure 8. The lattice fringes of the rGO (3.36°) and TiO_2_ (3.49°) correspond to the rGO (002) plane and TiO_2_ (101) plane, respectively [31].

### 3.3. AFM

The topologies and cross-section of rGO-TiO_2_ nanocomposite were analyzed by atomic force microscopy (AFM) as shown in Figure 6. Scanning areas for the surface were up to 300 nm × 300 nm whereas Figure 6a shows the 3D images with the highest depth of ~25 nm. In addition, the entire thickness with roughness of rGO-TiO_2_ nanocomposite was ~75 nm, which the highest with brightest color denoted as TiO_2_. Besides, the 2D image was focused under 250 nm scan area for a better view on the surface roughness (refer Figure 6b). It is clearly shown that there are two different formation colors, whereby the bottom with darker color classified as TiO_2_ was fully sputtered onto the rGO nanosheet and formed rGO-TiO_2_ nanocomposite whereas the standalone brighter color represented sputtered TiO_2_ that covered the top of the nanocomposite. Moreover, the cross-section of rGO-TiO_2_ nanocomposite along Figure 6b revealed that the surface thickness was ~35 nm, which was in agreement with Figure 6a.

### 3.4. FTIR

The functional groups of graphene oxide (GO) and rGO were completely analyzed and identified as shown in Figure 7. Several intense peaks appeared in the GO sample, indicating oxygen containing groups that successfully formed from graphite after oxidation (Figure 7a). The absorption peaks including aromatic C-H deformation at 670 cm^−1^, C-O stretching at 1052 cm^−1^, phenolic C-OH stretching at 1200 cm^−1^, C-OH at 1361 cm^−1^, hydroxyl groups of molecular water and C=C at 1625 cm^−1^, C=O stretching at 1729 cm^−1^, and a broad peak assigned as O-H stretching vibrations of C-OH groups at 3400 cm^−1^ [31]. Definitely, the broad band of O-H stretching at 3400 cm^−1^ is significantly reduced and also the presence of C-O at 1052 cm^−1^ and C-OH at 1361 cm^−1^ in the rGO pattern. These phenomena clearly indicate that the GO has been reduced and the oxygenated group is eliminated [38].

The FTIR transmission spectrum of rGO is placed in Figure 8 for further investigation between the TiO_2_ and rGO-TiO_2_ nanocomposite, which is formed via different sputtering temperatures. The FTIR spectrum of TiO_2_ was also been identified and depicted the peaks as high purity TiO_2_, which corresponded to TiO_2_. From the TiO_2_ spectrum, several peaks at 467 cm^−1^, 1345 cm^−1^, 1629 cm^−1^, and 3396 cm^−1^ can be observed. To the best of our understanding, the broad peak in the range of 500–1000 cm^−1^ region is ascribed to the Ti-O and Ti-O-Ti bridging stretching modes while the peak is denoted as anatase titania [39]. In the rGO-TiO_2_ samples, most of the rGO peaks did not appear in nanocomposite samples except 80 °C and 200 °C in the range between 1600–1750 cm^−1^, which indicated high C=C content. Interestingly, the intense peak absorption appeared for each rGO-TiO_2_ nanocomposite sample in the range of 550–900 cm^−1^ that was designated as Ti-O-C or Ti-O-Ti linkage bonds formed. This shows that these nanocomposite samples were well formed and established agreement for rGO-TiO_2_ via sputtering method [40].

### 3.5. XRD

The XRD pattern was utilized to analyze the crystallinity of introduced TiO_2_ that sputtered onto the rGO nanostructure. Figure 9 shows the XRD spectra for synthesized rGO, TiO_2_, and rGO-TiO_2_ nanocomposites deposited at various sputtering temperatures. The XRD of rGO had a sharp peak presence at 25.2°, 43.8°, and 45.6° as shown in Figure 9a. These peaks correspond to (002), (001), and (001) diffraction planes while the 25.2° peak indicated that the reduction process from graphene oxide (GO) was successfully obtained [41,42]. Moreover, less intense peaks at 43.8° and 45.6° indicated highly disordered carbon material [32]. On the other hand, Figure 9b shows XRD patterns of high crystallinity TiO_2_ as raw nanoparticle recorded in the range from 15° to 65°. The sharp Bragg peaks indicate that the highly crystalline TiO_2_ nanomaterials are well-formed. The presence of the broad peaks and Bragg diffraction peaks indexed along 25.4°, 28.1°, 41.0°, and 54.6° with (101), (112), (211), and (204) orientations, respectively, corresponded to anatase phase TiO_2_ (JCPDS card no: 21-1272) [43]. The XRD patterns from (c) to (g) were detected for rGO-TiO_2_ nanocomposite in variations of sputtering temperature and well aligned along at 25.2°, which were in good agreement with the obtained unique properties along the (101) orientation.

### 3.6. Raman

The Raman spectra of rGO, anatase TiO_2_, and rGO-TiO_2_ nanocomposites deposited at temperatures of 40 °C, 80 °C, 120 °C, 160 °C, and 200 °C, respectively, are shown in Figure 10. The appearance of rGO peaks in the Raman spectrum of D band and G band at 1348.20 cm^−1^ and 1592.84 cm^−1^, respectively, as analyzed in our previous work, confirmed successful reduction of GO to rGO [31,44]. Besides, the significance peaks for the sputtered TiO_2_ have been identified as anatase phase TiO_2_ due to the aligned Raman frequencies at 148.24 cm^−1^ (E_g1_), 391.37 cm^−1^ (B_1g_), 508.64 cm^−1^ (A_1g_), and 629.65 cm^−1^ (E_g_), which correspond to the literature [45]. Meanwhile, the Raman spectrum of rGO-TiO_2_ nanocomposites with different sputtered temperatures have recognized entire material characteristics as every essential peak for particular anatase TiO_2_ and rGO clearly appeared in the composite. Furthermore, the I_D_/I_G_ ratios of rGO and rGO-TiO_2_ nanocomposites were calculated and displayed in Figure 10. The I_D_/I_G_ ratio could determine the defects of carbon nanomaterial based on the intensity of D band and G band. Among these rGO-TiO_2_ nanocomposites, sputtered temperature condition at 80 °C was the highest defect credited to its I_D_/I_G_ ratio. In contrast, the sputtered temperature at 200 °C with lowest I_D_/I_G_ ratio indicated that the ideal sp^2^ C-C network was well formed. Based on our understanding, D band-mode represented disordered structure of graphene material (sp^3^-bonded) whereas G band arose from C-C bond stretching in graphitic material or known as more relevant to sp^2^-bonded carbon atoms [46].

### 3.7. DSSCs

The DSSCs performance of sputtered raw TiO_2_ and rGO-TiO_2_ nanocomposite with different sputtering temperature onto rGO nanosheets were tabulated in Figure 11. The values of Table 2 are calculated based on the results in Figure 11 by reference from the equations below:(1)$$FF=\frac{V_{mp}J_{mp}}{J_{sc}V_{oc}}$$
(2)$$\eta =\frac{J_{sc}V_{oc}FF}{P_{in}}$$
where *Jsc* = short circuit current; *Voc* = open circuit voltage; *Jmp* = maximum current; *Vmp* = maximum voltage; *FF* = fill factor; *Pin* = input power; and *η* = efficiency.

The PCE performance (*η*) of the DSSCs based on sputtering temperature studies of rGO-TiO_2_ were determined accordingly by the details of photovoltaic characteristics such of DSSCs as short circuit current (*Jsc*), open circuit voltage (*Voc*), maximum power current (*Jmp*), maximum power voltage (*Vmp*), and fill factor (*FF*) (Table 2). It was revealed that 120 °C rGO-TiO_2_ nanocomposites obtained an ideal PCE result of 7.27% with *Jsc* of 15.74 A/cm^2^, *Voc* of 0.70 V, *Jmp* of 12.16 A/cm^2^, *Vmp* of 0.60, and *FF* of 0.66. Among these rGO-TiO_2_ nanocomposite samples, 120 °C also achieved the highest *Voc,* which indicated the shifting energy band of sputtered TiO_2_ with effective transferring of the photoinjected electrons from excited electron into the conduction band [47]. Furthermore, this impact would definitely benefit the 120 °C rGO-TiO_2_ nanocomposite with the efficient electron lifetime and obtained the highest value of *FF*.

From Table 2, a sputtered raw anatase TiO_2_ was measured and attained the lowest PCE with 0.79% while 40 °C sputtering temperature of TiO_2_ has the lowest PCE of 1.27% due to its small amounts of TiO_2_ content reacts at photoanode element and difficult to absorb visible light from solar simulator [48]. The PCE is significantly increased from 40 °C with 1.27% to 80 °C and 120 °C with 3.74% and 7.27%, respectively. There was an estimate that improved by double according to the increases of sputtering temperature. In contrast, the PCE dramatically dropped from 7.27% to 2.92% (160 °C) and lastly to 1.50% (200 °C). This occurrence might be due to the excessive amount of TiO_2_ which act as recombination centers that lead to high resistance of photo-induced charge carriers flow through the outer circuit [49].

## 4. Conclusions

This work discussed the effects of sputtering temperature of TiO_2_ introduced onto rGO nanosheet and photoanode film for DSSCs PCE performance was accomplished. The sputtering route for rGO-TiO_2_ nanocomposite formation is an impressive and effective approach. In this study, the rGO nanosheet was applied to facilitate photoinduced charge carrier electron transport while reducing electron-hole recombination pairs, resulting in better PCE performance of doped TiO_2_. The uniform distribution of TiO_2_ was found at 120 °C sputtering temperature on the rGO nanosheet as demonstrated by FESEM images where the lattice of rGO, TiO_2_, and rGO-TiO_2_ nanocomposites were presented. Surface roughness of rGO-TiO_2_ nanocomposite was measured at ~35 nm. Crystallinity of TiO_2_ onto rGO nanosheets was analyzed and this confirmed that a mixture of TiO_2_ anatase phase was sputtered. Both symmetry modes of rGO and anatase TiO_2_ were presented on rGO-TiO_2_ nanocomposite samples for various sputtering temperatures. The presence of Ti-O-C bonds was confirmed by FTIR spectra, associated with the oxygenated functional groups as shown in GO and rGO, respectively. It was found that 120 °C sputtering temperature eventually enhanced the overall mobility of electron transport to the outer circuit. The TiO_2_ sputtered at 120 °C possessed the ideal PCE of 7.27%, five times better PCE than the 40 °C sputtering temperature. The results indicated that precise charge carrier loading concentration of TiO_2_ is applied to achieve great absorptivity of dyes and charge separation, thus it improves the overall transportation properties. In contrast, the formation of rGO-TiO_2_ nanocomposite at highest sputtering temperature with 200 °C acquired the lowest PCE performance of 1.50%, which is attributed to its excessive amounts of TiO_2_ that penetrated and performed higher electron-hole pair recombination centers.

## Figures and Tables

**Figure 1 molecules-25-04852-f001:**
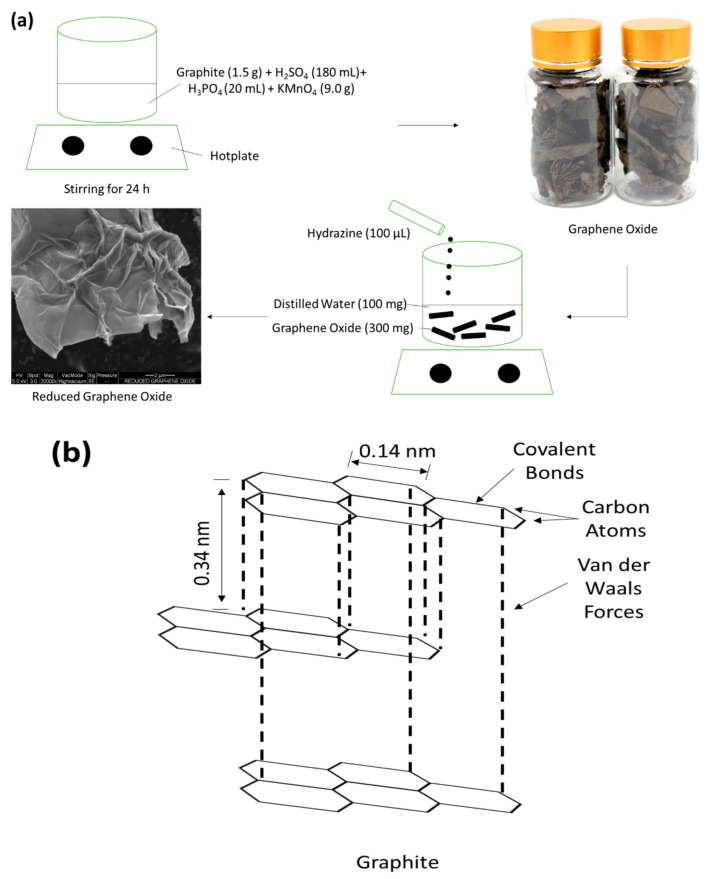
(**a**) Schematic diagram of GO and rGO synthesis, chemical structure of (**b**) graphite, (**c**) GO, and (**d**) rGO.

**Figure 2 molecules-25-04852-f002:**
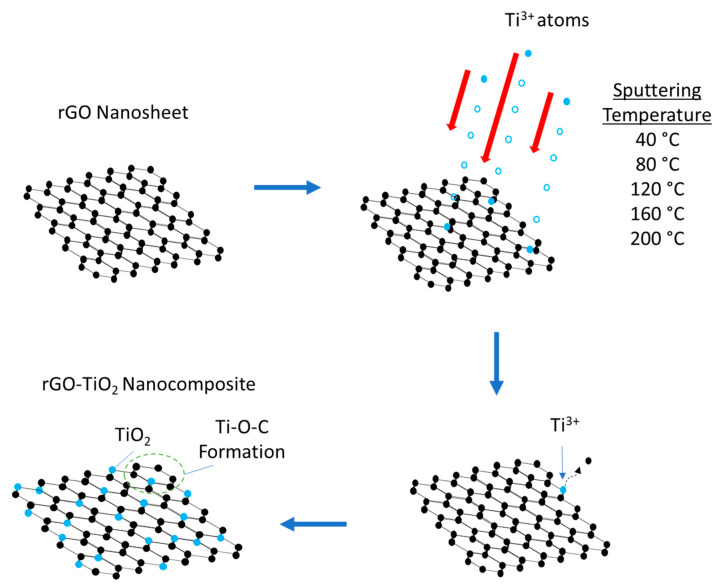
Sputtering mechanism process of rGO-TiO_2_ nanocomposite.

**Figure 3 molecules-25-04852-f003:**
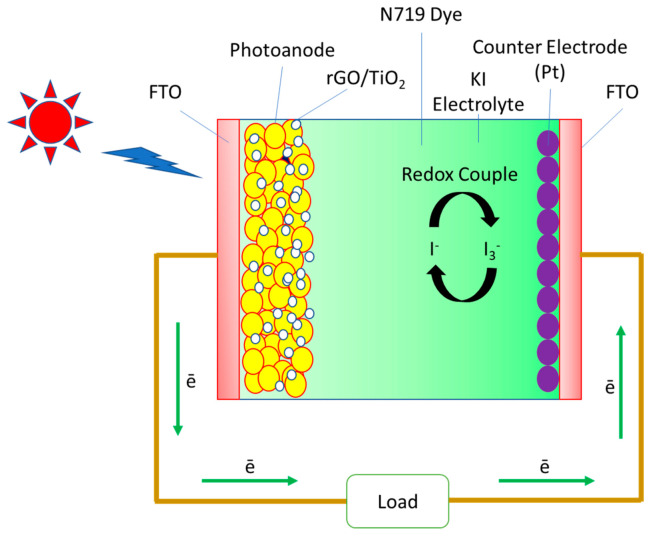
DSSCs schematic diagram of rGO-TiO_2_ nanocomposite photoanode.

**Figure 4 molecules-25-04852-f004:**
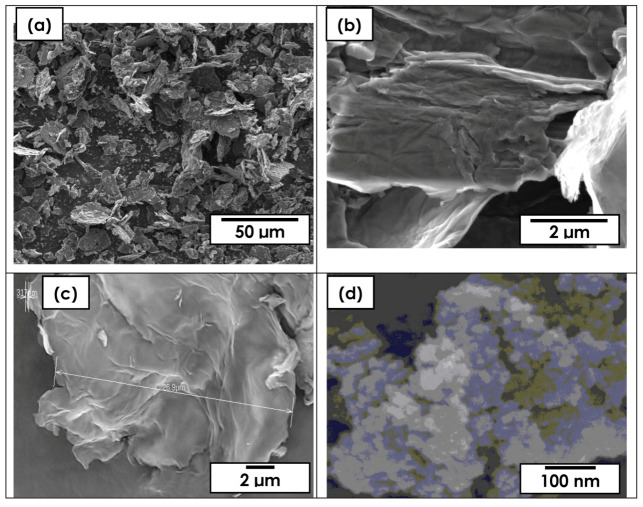
FESEM images of (**a**) graphite, (**b**) GO, (**c**) rGO, (**d**) TiO_2_, rGO-Ti ion implanted (**e**) at 120 °C, and (**f**) EDX data.

**Figure 5 molecules-25-04852-f005:**
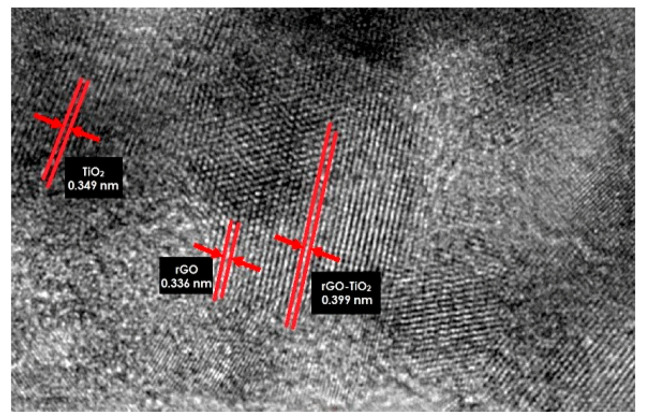
HRTEM image of 120 °C sputtered temperature of rGO-TiO_2_ nanocomposite.

**Figure 6 molecules-25-04852-f006:**
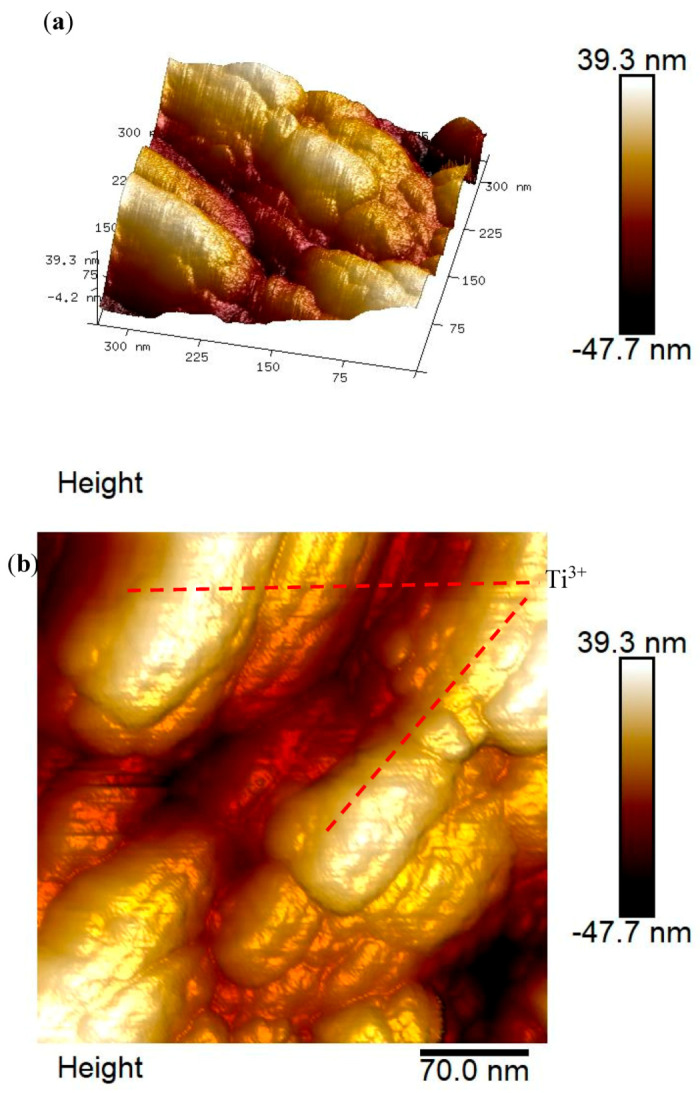
AFM images of 120 °C sputtered of rGO-TiO_2_ nanocomposite for (**a**) 3D orientation, (**b**) particular focus area, and (**c**) cross-section data.

**Figure 7 molecules-25-04852-f007:**
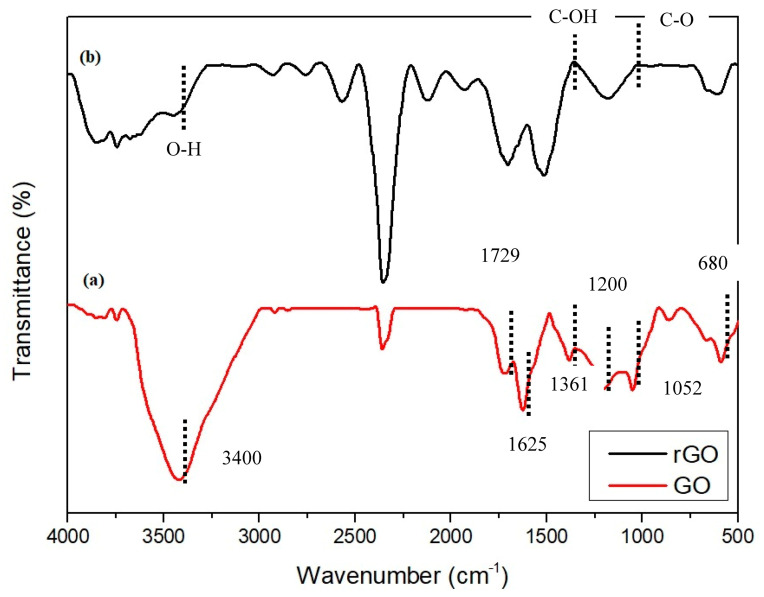
FTIR spectra of (**a**) GO and (**b**) rGO.

**Figure 8 molecules-25-04852-f008:**
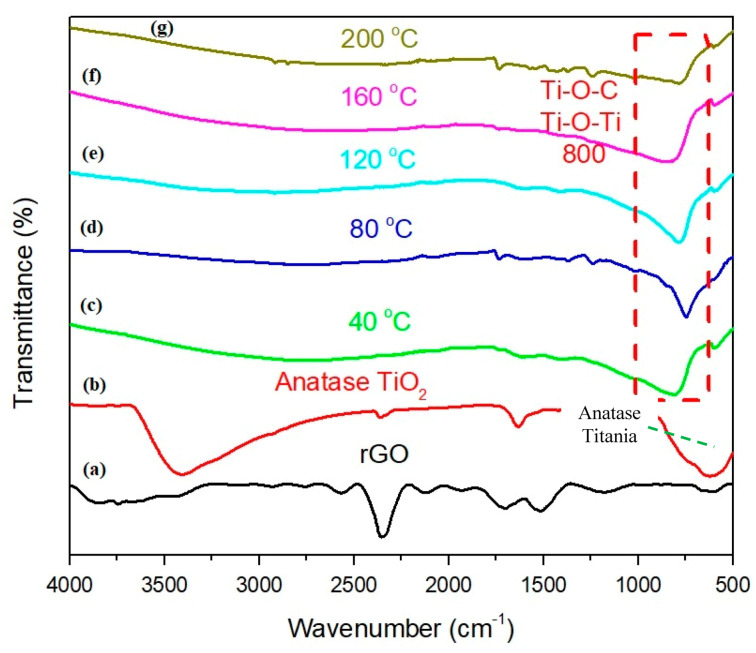
FTIR spectra of (**a**) rGO, (**b**) anatase TiO_2_, and rGO sputtered TiO_2_ via different temperature (**c**) 40 °C, (**d**) 80 °C, (**e**) 120 °C, (**f**) 160 °C, and (**g**) 200 °C.

**Figure 9 molecules-25-04852-f009:**
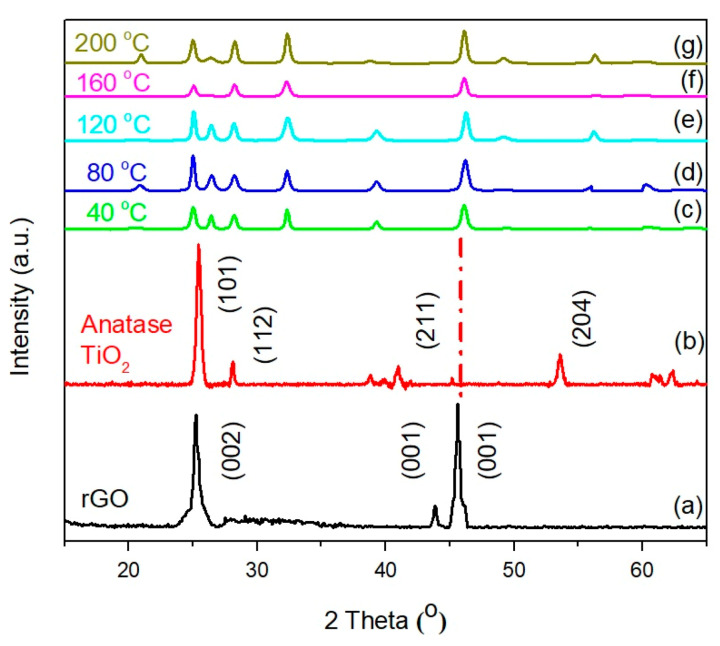
XRD patterns of (**a**) rGO, (**b**) anatase TiO_2_, rGO-TiO_2_ nanocomposite deposited at temperature of (**c**) 40 °C, (**d**) 80 °C, (**e**) 120 °C, (**f**) 160 °C, and (**g**) 200 °C.

**Figure 10 molecules-25-04852-f010:**
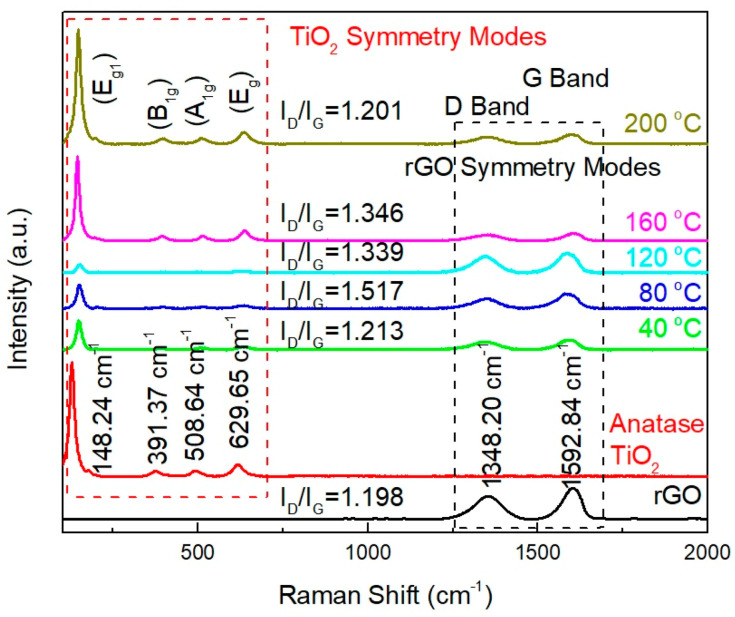
Raman spectra of rGO, anatase TiO_2_, and rGO-TiO_2_ nanocomposites based on different sputtered temperatures from 40 to 200 °C.

**Figure 11 molecules-25-04852-f011:**
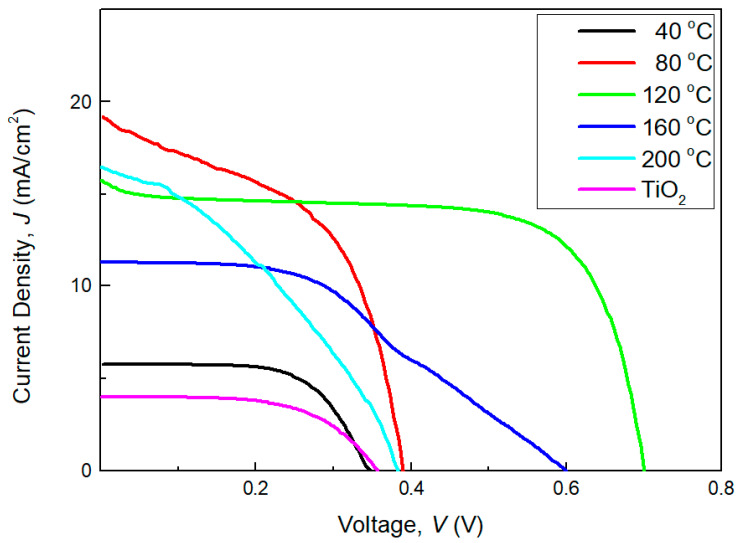
*J-V* curve of DSSCs based anatase TiO_2_ and various sputtering temperatures of rGO-TiO_2_ nanocomposite photoanodes film.

**Table 1 molecules-25-04852-t001:** Summary of rGO-TiO_2_ nanocomposites by different techniques.

Composite Formation	Optimized Concentration	Dopants	Method	Findings	Ref.
rGO/TiO_2_	3 wt%	GO	solvothermal	ACT degradation and mineralization on photocatalytic	[26]
TiO_2_-rGO	0.5 wt%	GO	hydrothermal	FM photodegradation	[27]
TiO_2_-rGO	0.4 wt%	GO	hydrothermal	DSSCs	[28]
rGO-TiO_2_	0.5mg	rGO	hydrothermal	DSSCs	[29]
Ag/rGO/TiO_2_	-	GO	solvothermal	Plasmonic DSSCs	[30]

**Table 2 molecules-25-04852-t002:** Summary of photovoltaic characteristics of DSSCs via different sputtering temperatures of TiO_2_.

Sputtering Temperature, °C	Short Circuit Current, *Jsc*	Open Circuit Current, *Voc*	Maximum Power Current, *Jmp*	Maximum Power Voltage, *Vmp*	Fill Factor, *FF*	Efficiency, *η*
TiO_2_	3.98	0.36	2.82	0.28	0.55	0.79
40	5.75	0.35	5.07	0.25	0.63	1.27
80	19.18	0.39	12.58	0.30	0.50	3.74
120	15.74	0.70	12.16	0.60	0.66	7.27
160	11.31	0.60	9.69	0.30	0.43	2.92
200	16.46	0.38	14.85	0.10	0.24	1.50

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
