# Peer review of "Influence of Sputtering Temperature of TiO2 Deposited onto Reduced Graphene Oxide Nanosheet as Efficient Photoanodes in Dye-Sensitized Solar Cells"

_molecules, 2020, doi:10.3390/molecules25204852_

Round 1

Reviewer 1 Report

Low et. al. report on the synthesis of reduced graphene oxide which was subsequently coated with TiO2 via RF sputtering with measures taken to improve the photogeneration and transport of charge through the rGO-TiO2. The rGO-TiO2 nanocomposites were characterized with XRD, FESEM, EDX, TEM, AFM, Raman, and FTIR. The performance of the fabricated was evaluated through current and voltage measurements.

Comments:

  • No information was given on the percent yield of the rGO produced.

  • Concentration of dye used the fabrication of solar cell was not reported

  • What is the purpose of characterizing the graphite?

  • Authors did not elaborately discuss the mechanism by which rGO enhance the efficiency or performance of dye sensitized solar cells

  • English language must be improved.

Author Response

RESPONSE TO REVIEWERS REPORT (Materials- 948693)

Thank you very much for your constructive suggestions in “materials- 948693 Decision Letter” on 22 September 2020. Based on reviewers’ comments and suggestions, the changes within the revised manuscript have been highlighted and our responses are as follows.

Kind Regards,

Dr. Low Foo Wah

Reviewer Comments:

Low et. al. report on the synthesis of reduced graphene oxide which was subsequently coated with TiO2 via RF sputtering with measures taken to improve the photogeneration and transport of charge through the rGO-TiO2. The rGO-TiO2 nanocomposites were characterized with XRD, FESEM, EDX, TEM, AFM, Raman, and FTIR. The performance of the fabricated was evaluated through current and voltage measurements.

  • No information was given on the percent yield of the rGO produced.
  • The yield percentage value is calculated and added in the methodology section.
  • Concentration of dye used the fabrication of solar cell was not reported
  • The concentration of N719 dye is included in the text.
  • What is the purpose of characterizing the graphite?
  • I think the graphite result only characterized and reported in morphology part. The graphite shown in morphology is for readers to compare the structural for before and after synthesized GO from the raw material.
  • Authors did not elaborately discuss the mechanism by which rGO enhance the efficiency or performance of dye sensitized solar cells
  • The mechanism of rGO for improve the efficiency is added in section 2.4. We didn’t add the rGO material as alone in DSSCs part is because the rGO is not metal oxide that enable to absorb light. It is only use for fasten the electron flow process or catalyst role. The rGO alone will shown in a flat results and close to 0% efficiency.
  • English language must be improved.
  • The manuscript is sent for professional English editing after revised.

Reviewer 2 Report

The manuscript ‘Influence of Sputtering Temperature of TiO2 Deposited onto Reduced Graphene Oxide Nanosheet as Efficient Photoanodes in Dye-Sensitized Solar Cells’ used the improved Hummer’s experimental methods to synthesize an ultrathin reduced graphene oxide nanosheet. They also done physical characters analysis and performance of DSSCS measurement to explore the its application in solar cells. In generally, the study has certain meaningful in the field of optoelectronic. But there are some questions that should be resolved, which is listed as following:

  1. in introduction, author should introduce the basic structure of Graphene Oxide Nanosheet and research development of rGO in DSSCs. As we known, Graphene as abundantly available carbon is viewed as one of the most promising materials in solar cells since first report, because of the characteristics of possesses high carrier mobility and low sheet resistance and so on. It can be designed by oxygen-containing functional groups, like −OH, =O, and −COOH, and lattice surface defects. And those designed methods will make influence on the performance of graphene-Based photoelectrode materials (DOI: 10.1155/2019/1812879), please add this discussion.
  2. The author mainly analyzed the microstructure of titanium dioxide after sputtering to rGO Nanosheet, but there was a lack of performance characterization of such composite materials, such as edge conduction band energy level, conductivity, light transmittance and so on.
  3. The author's application of this composite photoanode material in Section 3.7 of the results and discussion should be compared with traditional photoanode materials to highlight the advantages of this material.
  4. When analyzing SEM, the author should compare the images at different sputtering temperatures, so as to reflect the effect of temperature on sputtering more intuitively.
  5. What temperature is the TEM image taken by the author obtained?
  6. Why not analyze TEM images under other temperature conditions?
  7. Figure 10 should be revised to Figure 6 in the author's writing sequence.
  8. The same problem exists in AFM image analysis.
  9. In the analysis of XRD, the author can simply mark the information of each peak, so that the comparison can be more intuitive.
  10. The author must examine the manuscript throughout, such as Figure 1 (b) and references to make it clearly.

Author Response

RESPONSE TO REVIEWERS REPORT (Materials- 948693)

Thank you very much for your constructive suggestions in “materials- 948693 Decision Letter” on 27 September 2020. Based on reviewers’ comments and suggestions, the changes within the revised manuscript have been highlighted and our responses are as follows.

Kind Regards,

Dr. Low Foo Wah

Reviewer Comments:

The manuscript ‘Influence of Sputtering Temperature of TiO2 Deposited onto Reduced Graphene Oxide Nanosheet as Efficient Photoanodes in Dye-Sensitized Solar Cells’ used the improved Hummer’s experimental methods to synthesize an ultrathin reduced graphene oxide nanosheet. They also done physical characters analysis and performance of DSSCS measurement to explore the its application in solar cells. In generally, the study has certain meaningful in the field of optoelectronic. But there are some questions that should be resolved, which is listed as following:

  1. In introduction, author should introduce the basic structure of Graphene Oxide Nanosheet and research development of rGO in DSSCs. As we known, Graphene as abundantly available carbon is viewed as one of the most promising materials in solar cells since first report, because of the characteristics of possesses high carrier mobility and low sheet resistance and so on. It can be designed by oxygen-containing functional groups, like −OH, =O, and −COOH, and lattice surface defects. And those designed methods will make influence on the performance of graphene-Based photoelectrode materials (DOI: 10.1155/2019/1812879), please add this discussion.
  • The development graphene works in literature for DSSCs is further discuss in introduction section.
  1. The author mainly analyzed the microstructure of titanium dioxide after sputtering to rGO Nanosheet, but there was a lack of performance characterization of such composite materials, such as edge conduction band energy level, conductivity, light transmittance and so on.
  • Thank you for these recommendations, our team will further do the further testing based on your advice in next publication.
  1. The author's application of this composite photoanode material in Section 3.7 of the results and discussion should be compared with traditional photoanode materials to highlight the advantages of this material.
  • The raw material of sputtered TiO2 data is added into the DSSCs figure to made the comparison between precursor and decorated samples.
  1. When analyzing SEM, the author should compare the images at different sputtering temperatures, so as to reflect the effect of temperature on sputtering more intuitively.
  • We took the desired sample according to the DSSCs. In this SEM our objective is viewing the morphology of the material and currently we have limited source on this study.
  1. What temperature is the TEM image taken by the author obtained?
  • The HRTEM image is 120 °C sputtering temperature sample (the highest PCE) and the caption is revised.
  1. Why not analyze TEM images under other temperature conditions?
  • For this characterize result, our purpose is to view and calculate the lattice of each element that presence in this material.
  1. Figure 10 should be revised to Figure 6 in the author's writing sequence.
  • The figure 9 and figure 10 is moved after figure 6 (AFM).
  1. The same problem exists in AFM image analysis.
  • We took the desired samples for this characterization according to the DSSCs result.
  1. In the analysis of XRD, the author can simply mark the information of each peak, so that the comparison can be more intuitive.
  • The XRD figure is revised and added some information accordingly.
  1. The author must examine the manuscript throughout, such as Figure 1 (b) and references to make it clearly.
  • Yes, the overall content especially figure is revised and the English is sent to editing professional groups.

Round 2

Reviewer 2 Report

The manuscript report a study of Influence of Sputtering Temperature of TiO2 Deposited onto Reduced Graphene Oxide Nanosheet as Efficient Photoanodes in Dye-Sensitized Solar Cells. This paper takes graphene&TiO2 as the research object in the utility of DSSCs, and the results provide some guidance for the use of clean energy. The author has revised the article, however, there are still some problems to be resolved, which is listed as follows:

1.There have also some English express error that should be checked and revised:

In abstract: a) ‘incorporation’with should be ‘incorporating’; b)‘resulting undesirable charge’ should be ‘resulting in undesirable charge’;

Introduction: a) ‘The demand of global energy’ should be ‘The demand for global energy’; b)‘resulting undesirable charge’ should be ‘resulting in undesirable charge’;c)

The sentence ‘Eventually, the sputtering technique is promising as to maximize the opportunity of filling the oxygen vacancy in order to reduce the intrinsic defect of rGO in oxides lattice with the presence of TiO2.’can be revised as ‘Eventually, the sputtering technique is promising to maximize the opportunity to fill the oxygen vacancy to reduce the intrinsic defect of rGO in oxides lattice with TiO2’ï¼›

In conclusion: the sentence is unclear ‘In this study, the rGO nanosheet was applied for facilitating photoinduced charge carrier electron transport while reducing electron-holes recombination pairs, resulting better PCE performance of doped TiO2, which can be revised as ‘In this study, the rGO nanosheet was applied to facilitate photoinduced charge carrier electron transport while reducing electron-holes recombination pairs, resulting in better PCE performance of doped TiO2.’

2.Figure 4f. the inserted data table is not clear. Please modify this graph; at the same time, Figure 6.a inserted AFM images is not clear, including coordinates and data.

3. The format of references should be unified. Please check the use of the author's full name, abbreviation and ‘et al’. Please unify the whole References.

Author Response

RESPONSE TO REVIEWERS REPORT (Materials- 948693)

Thank you very much for your constructive suggestions in “materials- 948693 Decision Letter” on 9 October 2020. Based on reviewers’ comments and suggestions, the changes within the revised manuscript have been highlighted and our responses are as follows.

Kind Regards,

Dr. Low Foo Wah

Reviewer Comments:

The manuscript report a study of Influence of Sputtering Temperature of TiO2 Deposited onto Reduced Graphene Oxide Nanosheet as Efficient Photoanodes in Dye-Sensitized Solar Cells. This paper takes graphene&TiO2 as the research object in the utility of DSSCs, and the results provide some guidance for the use of clean energy. The author has revised the article, however, there are still some problems to be resolved, which is listed as follows:

1.There have also some English express error that should be checked and revised:

In abstract: a) ‘incorporation’with should be ‘incorporating’; b)‘resulting undesirable charge’ should be ‘resulting in undesirable charge’;

Introduction: a) ‘The demand of global energy’ should be ‘The demand for global energy’; b)‘resulting undesirable charge’ should be ‘resulting in undesirable charge’;c)

The sentence ‘Eventually, the sputtering technique is promising as to maximize the opportunity of filling the oxygen vacancy in order to reduce the intrinsic defect of rGO in oxides lattice with the presence of TiO2.’can be revised as ‘Eventually, the sputtering technique is promising to maximize the opportunity to fill the oxygen vacancy to reduce the intrinsic defect of rGO in oxides lattice with TiO2’ï¼›

In conclusion: the sentence is unclear ‘In this study, the rGO nanosheet was applied for facilitating photoinduced charge carrier electron transport while reducing electron-holes recombination pairs, resulting better PCE performance of doped TiO2, which can be revised as ‘In this study, the rGO nanosheet was applied to facilitate photoinduced charge carrier electron transport while reducing electron-holes recombination pairs, resulting in better PCE performance of doped TiO2.’

- The suggestion of statement is revised accordingly.

2.Figure 4f. the inserted data table is not clear. Please modify this graph; at the same time, Figure 6.a inserted AFM images is not clear, including coordinates and data.

- The figures are revised accordingly.

  1. The format of references should be unified. Please check the use of the author's full name, abbreviation and ‘et al’. Please unify the whole References.

- The format of references is revised accordingly.
